# Study on the Interactions of Cyclins with CDKs Involved in Auxin Signal during Leaf Development by WGCNA in *Populus alba*

**DOI:** 10.3390/ijms241713445

**Published:** 2023-08-30

**Authors:** Jinghui Liang, Zhaoqun Wu, Xiusheng Zhang, Xin Du, Shiyi Wang, Yebo Yang, Yuwen Wang, Yiming Wang, Hailing Yang

**Affiliations:** State Key Laboratory of Tree Genetics and Breeding, The Tree and Ornamental Plant Breeding and Biotechnology Laboratory of National Forestry and Grassland Administration, College of Biological Sciences and Technology, Beijing Forestry University, Beijing 100107, China; liangjinghui@bjfu.edu.cn (J.L.); qunzhaowu@bjfu.edu.cn (Z.W.); zxx666@bjfu.edu.cn (X.Z.); duxin_2020@bjfu.edu.cn (X.D.); wangshiyi@bjfu.edu.cn (S.W.); yangyebo@bjfu.edu.cn (Y.Y.); wyw1216558622@bjfu.edu.cn (Y.W.); wangyiming@novogene.com (Y.W.)

**Keywords:** cyclin, CDK, auxin, WGCNA, Y2H, leaf development, *Populus alba*

## Abstract

Cell division plays an indispensable role in leaf morphogenesis, which is regulated via the complexes formed by cyclin and cyclin-dependent kinase (CDK). In this study, gene family analysis, exogenous auxin stimulation, RNA-seq and WGCNA analysis were all used to investigate the molecular mechanisms by which cell-cycle-related factors participated in the auxin signaling pathway on leaf morphogenesis. Sixty-three cyclin members and seventeen CDK members in *Populus alba* were identified and systematically analyzed. During the evolution, WGD was the main reason that resulted in the expansion of *cyclin* and *CDK* genes. Firstly, after a short time treating with auxin to matured leaves of seedlings, genes related to cell division including *GRF* and *ARGOS* were both upregulated to restart the transition of cells from G1-to-S phase. Secondly, with three days of continuous auxin stimulation to leaves at different developmental stages, leaves area variation, transcriptomes and hormones were analyzed. By PCA, PCoA and WGCNA analyses, the turquoise module was both positively related to leaf development and auxin. Based on the co-expression analysis and Y2H experiment, PoalbCYCD1;4, PoalbCYCD3;3 and PoalbCYCD3;5 were supposed to interact with PoalbCDKA;1, which could be the trigger to promote the G1-to-S phase transition. The ARF transcription factor might play the key role of connecting the auxin signaling pathway and cell division in leaf morphogenesis by affecting CYC–CDK complexes.

## 1. Introduction

The *Populus alba* is a fast-growing tree and a considerable biomass product. Moreover, it is widely distributed in Europe and Asia, with excellent resistance to insect pests, germs, drought, salt and low-temperatures stresses [1]. The rapid and healthy growth of poplar needs a large quantity of energy derived from the leaves [2]. Because of the assembly genome of *P. alba*, it is meticulous, systematic and convenient to do the research on tree leaf morphogenesis [3].

Leaf originates from primordium which occurs in the area of auxin concentration in the shoot apex meristem (SAM) [4]. Then, polarities establish and leaf intercalary growth happens [5]. Leaf development involves cell division [6]. When the leaf is young, most cells are in the stage of proliferation. With the continuous growth, cell division will stop from the tip of the leaf, and cell expansion will begin. The dividing line between the division area and the expansion area is called “arrest front”. When the arrest front moves from the tip to the bottom of leaf, most cells are in the stage of expansion [7,8].

The leaf cell division involves cyclins and cyclin-dependent kinases (CDKs). They are both key regulators which can form complexes to stimulate cells through the G1-to-S and G2-to-M checkpoints [9]. They play a crucial and beneficial role in plant development. In *Arabidopsis thaliana*, *CYCA2;1* and *CYCA2;4* could promote the vein cell proliferation [10]. The elevated expression of *CYCB1;1* might cause the continuity of cell division and lead to the persistent leaf marginal growth [11]. In *Oryza sativa*, *OsCYCD3;1* could maintain meristem activity and regulate branch formation [12]. Overexpression of *CYCD1;2* resulted in increasing of cell number and alteration of leaf morphology in a hybrid poplar [13]. Ectopic expression of *PsnCYCD1;1* in *Nicotiana tabacum* led to leaf curl and a greater number of epidermal cells. Additionally, cell-cycle-related genes, including *NtCDKA*, *NtCDKB*, and *NtE2F*, stem-development-related genes, including *NtSTM* and *NtKNAT1*, as well as *NtAS*, related to the leaf polarity, were all upregulated [14]. Overexpression of *CYCD3;3* in poplar led to larger and wrinkled leaves. A thickened stem and more branches also appeared [15].

Auxin also plays a positive role in leaf development. Auxin biosynthesis and cellular efflux collaborated in regulating leaf vein patterning [16]. A proximodistal growth gradient that improved lateral growth, and consequently, the characteristic ellipsoid *A. thaliana* leaf shape was organized by WOX-mediated auxin biosynthesis [17]. A stable boundary between adaxial and abaxial cell fates is specified several plastochrons before primordium emergence when high auxin levels accumulate on a meristem prepattern [18]. Additionally, many references revealed that auxin could affect cyclins and CDKs. When *IAA28* was removed by *miR847*, both *CYCD3;1* and *CYCB1;1* were upregulated with improvement of auxin signaling, which enhanced meristem ability and extended the development time [19]. In “84K” poplar, the expression of cyclin was increased with *miR393* blocked and the auxin signaling pathway was enhanced [20]. Overexpression of *TaKLU* increased the accumulation of auxin in leaves. The *CDKA;1*, *CYCT1;1* and *CDC20* (*CELL DIVISION CYCLE 20*) were upregulated with promotion of cell division and prolongation of leaf expansion time [21]. In BY-2 suspension cells, auxin was found to be necessary to activate the formation of CDKA-related complexes [22].

Based on the previous studies, we speculated that cell-cycle-related factors, auxin and leaf development are closely related. However, the key cyclins and CDKs activating the poplar leaf cell proliferation were still unidentified. Moreover, the potential mechanism by which the auxin influences the cell-cycle-related factors remained unknown. The construction of their relationship network was critical in understanding the leaf growth. Thus, we proposed a hypothesis that implicates cyclins and CDKs in the auxin signaling pathway, resulting in cell division and contributing to leaf morphogenesis.

Due to the large size and slow growth, classical genetic screens of mutants with distinct developmental phenotypes are difficult for tree study. Hence, the information on leaf growth can be obtained more effectively through physiological and transcriptomic analyses [23]. The genome-wide identification and analysis of gene families, RNA-seq, hormone measurement, PCA analysis and WGCNA analysis were all used for the study of poplar leaf morphogenesis. The interaction relationship of critical cyclins and CDKs would be identified and a mechanism by which auxin signaling pathway influenced cell cycle process would be conjectured. Our research further optimized the molecular mechanism of poplar leaf development and facilitated the rapid and healthy poplar cultivation.

## 2. Results

### 2.1. Identification of the Cyclin and CDK Gene Families in P. alba

The candidate cyclin members were preliminarily screened from *P. alba* genome by hidden Markov model (HMM) profiles of the cyclin_N domain (PF00134) and cyclin_C domain (PF02984). The cyclin_N domain is more conserved than the cyclin_C domain [24]. Consequently, some sequences were removed due to the severely incomplete cyclin_N domain. Finally, 63 cyclin members in *P. alba* were identified (Appendix A). After BLASTP, filtering the CKL members and confirming the CDK domain, 17 CDK members in *P. alba* were identified (Appendix A).

### 2.2. Phylogenetic Analysis and Classification

It was reported that at least 49 cyclins were discovered in *A. thaliana*, which were divided into 10 types [24]. To investigate the evolutionary relationships and classification of cyclins in *P. alba*, we built the phylogenetic tree using PhyML software (Figure 1A). According to the phylogenetic tree, 63 cyclins in *P. alba* were divided into nine types from A to D, H, L, T, U and SDS. There were 9, 8, 2, 22, 1, 1, 6, 13 and 1 members in A to D, H, L, T, U and SDS types, respectively. However, J18 group member in *P. alba* was not found. A type was close to B type. Additionally, a small independent clade formed by C, H and L types was close to T type. As the largest group, D-type cyclins included 22 members. Moreover, U type was relatively independent of other types.

To verify and classify CDK family in *P. alba*, CKL members in *A. thaliana* were considered when building the phylogenetic tree (Figure 1B). Phylogenetically, 17 CDK members in *P. alba* were divided into seven groups from A to G. There were 1, 4, 2, 2, 2, 1 and 5 members in groups A to G, respectively. Group A was close to group B, while group C was close to the CKL. Other groups were relatively separated.

### 2.3. Sequences Analysis

Previous studies have shown that cyclin core exists in all cyclins, and it can be divided into two domains: cyclin_N and cyclin_C [9]. The cyclin_N domain is critical and highly conserved, whereas cyclin_C domain is less conserved and not present in all cyclins [24]. After identification and classification, we found that cyclin_N domain existed in all cyclin members, while cyclin_C was absent from C, L, U and 5 of 6 T-type cyclins. Pkinase domain and PK_Tyr_Ser-Thr domain existed in all identified CDK members (Appendix A).

To investigate more information about cyclins and CDKs in *P. alba*, molecular weight (MW) and isoelectric point (pI) were calculated by ExPASY website (Appendix A). The lengths of 63 cyclins were ranging from 128 to 683 aa with MW and pI ranging from 14.69 to 77.93 kDa and 4.75 to 9.80. The lengths of 17 CDKs were ranging from 146 to 822 aa with MW and pI ranging from 16.50 to 93.23 kDa and 4.45 to 9.52.

Analysis of exon/intron distribution of *cyclin* and *CDK* gene families can help to understand their respective functions and evolutionary relationships. Consequently, the number and distribution of exons of *cyclin* and *CDK* gene families were analyzed (Appendix A). The *cyclin* gene with the largest number of exons (22) was *PoalbCYCB3;1*. The average number of exons included in group A to D, H, L, T, U and SDS of *cyclin* genes were 11.56, 11.38, 7.00, 5.45, 13.00, 9.00, 7.00, 2.08 and 8.00, respectively. Additionally, 12 of 13 U-type *cyclin* genes had 2 exons, which suggested that their gene structures were highly conserved. Moreover, the *CDK* gene with the largest number of exons (13) was *PoalbCDKC;2*, while *PoalbCDKE;1* had only one exon. The average number of exons included in group A to G of *CDK* genes were 7.00, 6.00, 11.50, 8.50, 1.50, 2.00 and 6.20.

To illustrate the conserved motifs in cyclins and CDKs, 10 type motifs were predicted by MEME website (Appendix A). Motif 1 existed in all cyclins, which suggested that it is a critical motif for cyclins. Motif 6, 8 and 10 were only identified in A- and B-type cyclins. Motif 9 specifically existed in D-type cyclins, while motif 2 and 7 only existed in U-type cyclins. Motif 3, 4 and 5 were commonly identified in all types of cyclins except U type. The research on CDKs motifs suggested that 6 motifs were identified in CDKA, while both group B and D shared eight common motifs. There were four motifs which were found in PoalbCDKG;1 due to the short length, while another four members in group CDKG shared 10 common motifs. Additionally, at least six motifs were identified and distributed in group C, E and F. This evidence indicated that CDKs in *P. alba* were highly conserved. In general, our results showed that the cyclins and CDKs in the same group contained the parallel motif types.

### 2.4. Chromosomal Distribution and Duplicated Gene Pairs

Chromosomal distribution of *cyclin* and *CDK* genes were determined based on the information from the *P. alba* genomic database (Appendix A). The results revealed that 63 *cyclin* genes were mapped onto 18 chromosomes, while 17 *CDK* genes were distributed on 13 chromosomes (Figure 2). Although 18 chromosomes contained *cyclin* or *CDK* genes, the overall distribution was mostly uneven. Chromosomes 2 and 5 both contained eight *cyclin* genes, while only one *cyclin* gene was distributed on chromosome 4. Moreover, chromosome 12 had three members of *CDK* genes, while chromosome 3 contained only one *CDK* gene. Interestingly, chromosome 17 did not contain any *cyclin* or *CDK* genes.

Duplicated genes in plants were attributed to ancient duplication events [25]. Consequently, DupGen_finder software was employed to find out the duplicated *cyclin* and *CDK* gene pairs in *P. alba* (Appendix A). The colorful lines in circos figure represented the duplicated *cyclin* and *CDK* gene pairs (Figure 2). There were 48 *cyclin* duplicated gene pairs and 11 *CDK* duplicated gene pairs, and most of them were involved in WGD events. Additionally, the duplication type of three *cyclin* gene pairs (*PoalbCYCA3;2/PoalbCYCA3;3*, *PoalbCYCC1;1/PoalbCYCT1;4*, *PoalbCYCC1;2/PoalbCYCT1;4*) were regarded as transposed duplication, while only one *cyclin* duplicated gene pair (*PoalbCYCU4;1/PoalbCYCU4;2*) was considered as proximal type genes. Moreover, 4 of 11 *CDK* duplicated gene pairs (*PoalbCDKA;1/PoalbCDKB1;2*, *PoalbCDKC;1/PoalbCDKC;2*, *PoalbCDKD;1/PoalbCDKD;3*, *PoalbCDKG;1/PoalbCDKG;3*) were identified as transposed type genes. However, tandem duplication was not found in *cyclin* and *CDK* genes in *P. alba*. All this evidence illustrated that whole genome duplication (WGD) was the main way that resulted in the expansion of *cyclin* and *CDK* genes in *P. alba*. Furthermore, transposed, and proximal duplication also played important roles during the evolution. Interestingly, each of the five *CYCD3* genes in *P. alba* was collinear with the other four genes.

To explore the driving force for the evolutionary status of *cyclin* and *CDK* gene families, nonsynonymous (Ka) and synonymous (Ks) substitution rates were calculated by KaKs_calculator 2.0 (Appendix A). The results revealed that all Ka/Ks ratios of all *cyclin* and *CDK* pairs were less than 1, which indicated that purifying selection was the major driving force for the evolution of *cyclin* and *CDK* gene families in *P. alba*. Due to the same protein sequences of *PoalbCYCU4;1* and *PoalbCYCU4;2*, the Ka and Ks could not be calculated.

### 2.5. Promoter Analysis

Transcription factors can recognize the *cis*-acting elements in promoters to regulate the expression of the downstream genes [26]. Therefore, we investigated the 2000 bp upstream sequences of *cyclin* and *CDK* genes by PlantCARE website (Appendix A). The various *cis*-acting elements mainly involved in phytohormonal response, plant growth and development, and abiotic and biotic stress (Appendix A). AUXREs, TGA-element and AuxRR-core all responded to auxin, while CGTCA-motif and TGACG-motif both responded to MeJA. Besides, the results that GARE-motif, TATC-box and P-box appeared on the promoters of many *cyclin* and *CDK* genes suggested that gibberellin might affect the expression of these genes. In addition, the promoters of *PoalbCYCA2;3*, *PoalbCYCB2;3*, *PoalbCYCD3;1*, *PoalbCYCU2;2* and *PoalbCDKG;1* containing at least three TCA-elements revealed that salicylic acid might influence these genes. Moreover, ABRE elements were commonly found, which indicated that ABA might bring great impact on *cyclin* and *CDK* genes due to the existence of many ABRE elements. Additionally, MSA-like, A-box, O2-site, CAT-box and MBSI responded to cell cycle regulation, meristem activation, zein metabolism regulation, meristem expression and flavonoid biosynthesis, respectively. These elements were ubiquitous in the promoters of some *cyclin* and *CDK* genes. Furthermore, ARE, W-box, TC-rich repeats, G-box, MBS and LTR *cis*-acting elements involved in anaerobic induction, disease resistance, defense and stress response, salt resistance, drought inducibility and low-temperature response, respectively.

### 2.6. Interaction Prediction of Cyclin and CDK Proteins

Predicted interaction network of high confidence was constructed via the STRING online tool (Appendix A). It was obviously that PoalbCDKA;1 was in the center of the network, which might directly interact with A, B, H, D, U and SDS-type cyclins. PoalbCYCH1;1 might interact with PoalbCDKD and PoalbCDKF. PoalbCDKB2;2 might interact with PoalbCYCA and PoalbCYCB. Interestingly, PoalbCDKC and PoalbCYCT were isolated from the large network and formed a tiny system.

### 2.7. Leaf Responded to Short Time Treatment of Auxin at the Molecular Level

Recently, it has become evident that auxin plays a central role in leaf development [17]. In the first step of the experiment, leaves were picked up for RNA-seq after short-period treatment of auxin (Appendix A). After calculating the TPM, the differentially expressed genes (DEGs) (fold change > 1.5 and FDR < 0.05) were used for GO and KEGG enrichment. Compared with the control group, GO enrichment analysis revealed that the DEGs were mainly concentrated in cellulose synthesis, cell wall organization, cutin synthesis, secondary metabolite synthesis, carbohydrate biosynthesis and oxidoreductase activity. A total of 22 KEGG pathways were enriched, which suggested that DEGs were significantly enriched in brassinosteroids synthesis, secondary metabolite synthesis, plant hormone signal transduction, MAPK signaling pathway, peroxisome, photosynthesis, and protein transport (Appendix A).

After being treated with IBA, the genes related to the cell wall, including xyloglucan endotransglucosylase (*Poalb10G008630.1*) and pectinesterase inhibitors (*Poalb03G009440.1* and *Poalb15G002730.1*), were all upregulated. The elevated expressions of histone acetyltransferase (*Poalb07G001710.1*) with a 2.0-fold change and growth regulatory factor (*GRF*) (*Poalb18G000510.1*) with a 3.7-fold change were discovered. The expressions of Auxin-Regulated Gene involved in Organ Size (*ARGOS*) (*Poalb07G002120.1*), mitogen-activated protein kinase (*MAPK*) (*Poalb01G024420.1* and *Poalb05G009870.1*) and MAPK kinase (*MAPKK*) (*Poalb15G007140.1*) were all improved. The *PoalbCDKA;1*, *PoalbCDKF;1* and *PoalbCDKG;4* were all slightly upregulated. All this evidence revealed that the cells might be vigorous. Auxin-binding proteins (*ABP*) (*Poalb01G015200.1* and *Poalb06G022290.1*) and transmembrane kinase (*TMK*) (*Poalb16G010340.1*) were all downregulated, while the auxin signaling inhibitors (*AUX/IAA*) (*Poalb06G007640.1*, *Poalb01G016010.1* and *Poalb18G005650.1*) were upregulated. *GH3* (*Poalb09G008320.1* and *Poalb02G013890.1*) also showed the rising expression with a 2.5-fold change and 11.3-fold change, respectively. The expressions of ribonucleoside diphosphate reductase (*Poalb05G006800.1*) with a 1.7-fold change and DNA helicase (*Poalb14G001070.1*) with a 3.0-fold change were both rising. In addition, the expressions of *PoalbCYCD1;1* with a−1.6-fold change and the gene associated with DNA methylation (*Poalb05G017220.1*) with a −2.0-fold change were both reduced, while the expressions of S-phase related F-box proteins (*Poalb05G008660.1*, *Poalb16G010250.1*, *Poalb09G008920.1* and *Poalb06G002480.1*) were uplifted.

### 2.8. The Growth Curve of the Leaf Area from Young Leaves to Mature Leaves with Three Days Continuous Auxin Stimulation

Based on the first stage results, in the second step of the experiment, we analyzed leaf area changes, transcriptomes and hormone contents of leaves at different developmental stages under three days continuous auxin stimulation. After 3 days of growth, leaves were picked up at 72 h (Figure 3A,B). We recorded the leaf area at 0, 24, 48 and 72 h (Figure 3C and Appendix A). During the 0 to 72 h, the average area of the first to the fifth leaves expanded from 130 to 835 mm^2^, 452 to 1859 mm^2^, 1196 to 2963 mm^2^, 2452 to 3703 mm^2^ and 3405 to 4295 mm^2^, respectively (Figure 3D). Almost all the groups in L1, L2 and L3 all showed clear significant differences. Few significant differences were displayed in L4 and L5. Within 72 h, the growth multiples of area from the first leaf to the fifth leaf were 6.40, 4.11, 1.89, 1.51 and 1.26, respectively (Figure 3E). The univariate curve fitting was performed based on the changes in leaf area. The first through third leaves exhibited a substantial fitting effect, while the fitting curve for the fourth and fifth leaves had less explanatory power (Appendix A). These analyses indicated that the area of the first and second leaves grew rapidly, while the area of the third leaf to the fifth leaf grew more and more slowly.

### 2.9. Hormone Measurement and PCA Analysis

The contents of various hormones in leaves, including salicylic acid (SA), jasmonic acid (JA), ethylene synthesis precursor (ACC), indoleacetic acid (IAA), zeatin nucleoside (TZR), 12-oxy-phytodienoic acid (OPDA), gibberellin 1 (GA1), gibberellin 3 (GA3) and gibberellin 4 (GA4), were measured (Appendix A). IAA, SA, ACC and GA4 showed the highest content in L1 or L2 and the lowest content in L4 or L5, which displayed the downward trend from young leaves to the mature leaves on the whole. GA1 reached the minimum content of 32.14 ng/g in L1 and the maximum content of 52.64 ng/g in L5. The climbing trend of GA1 from L1-L5 was unique in these hormones. The content of GA3 initially increased from L1, reaching its peak in L3 about 20 ng/g, and subsequently declined until L5. The content of JA remained stable around 284 ng/g. In the principal component analysis (PCA), the first principal component axis (PC1) explained 60.59% of the total variation (Appendix A). The PCA plot showed that the hormone differences among the five groups were obvious. In contribution plot of PCA, SA and IAA made considerable contributions, while ACC and GA4 also accounted for proportions (Appendix A).

### 2.10. PCoA Analysis of Transcriptomes

To examine dynamic changes of transcriptomes during leaf development, transcripts per kilobase of exon model per million mapped reads (TPM) of all genes were used for principal coordinate analysis (PCoA) according to the global similarities/differences of samples (Appendix A). The PCoA plot showed that large dissimilarities existed among five groups and the global patterns of the transcriptomes varied significantly during the leaf development, as indicated to the first principal coordinate axis (PCoA 1), which explained 88.72% of the total variation (Appendix A). The second principal coordinate axis, which explained the variation among the in vivo transcriptomes, covered only 10.33% of the total variation. Based on the PCoA analysis, there was few differences within the groups and it was changes in leaf developmental stages that mainly caused obvious dynamic transcriptomes variation.

### 2.11. Construction of the Co-Expression Network

We used the TPM expression of all genes to construct a co-expression network, and combined this with the leaf area and the contents of multiple hormones. In the sample clustering tree, the differences from the first leaf to the fifth leaf gradually became larger (Figure 4A). Moreover, most hormones were distributed differently in various samples. After calculating the soft threshold, we constructed the co-expression network and modules by a one-step method. The module to which a gene belongs was determined by the absolute value of the eigengene connectivity (Appendix A). Nearly 31,000 genes were divided into 20 modules (Figure 4B and Appendix A). Turquoise module was the largest module with 15,906 genes, followed by the blue module with 5602 genes. The gene numbers of the most modules were basically maintained in hundreds. According to the correlation diagram between modules and samples, the turquoise module was generally positively correlated in the three samples of L1 with a correlation coefficient of 0.48 and a *p*-value of 0.07, but negatively correlated in the six samples of L4 and L5, with a correlation coefficient of −0.22. The blue module showed the contrary situation. It represented the negative correlation in six samples of L1 and L2 with an approximate correlation coefficient of −0.24, but the positive correlation in three samples of L5 with a correlation coefficient of 0.43 and a *p*-value of 0.1. Because the three biological replicates showed the high similarities in PCoA analysis, some modules that could not be correlated with all three biological replicates would not be considered. In the gene expression diagram of modules in samples, obviously, the genes expression level of the turquoise module was decreased gradually from L1 to L5, while the blue module was opposite to turquoise module (Figure 4D,E). Therefore, all this evidence manifested that the turquoise module was positively correlated with leaf development, while the blue module was negatively correlated with it.

### 2.12. GO and KEGG Analysis of Turquoise and Blue Modules

GO and KEGG analysis were applied to the genes in the turquoise and blue modules. The GO enrichment analysis revealed that the genes in the turquoise module were significantly enriched in the cell cycle, DNA replication, ribosome, protein translation, microtubules, vesicles, glycosylation, ATP metabolism and GTP binding (Appendix A). Moreover, the KEGG enrichment analysis indicated that the genes in the turquoise module was mainly enriched in the DNA replication, ribosome, amino acid synthesis, oxidative phosphorylation, proteasome, TCA cycle and fatty acid synthesis (Appendix A). The GO enrichment of the genes in the blue module was mainly concentrated in ion transport, autophagy, water shortage response, cold response, leaf senescence, protein catabolism, ubiquitin ligase and transmembrane transporters (Appendix A). KEGG enrichment of the genes in the blue module was significantly focused on autophagy, plant–pathogen interaction, ubiquitin mediated proteolysis, peroxisome and phosphatidyl inositol signaling systems (Appendix A).

### 2.13. Genes in the Turquoise and Blue Modules

The turquoise module included 70% *cyclins* and 47% *CDKs*. Additionally, *KNOLLE* (*Poalb13G010110.1*) and *PCNA* (*Poalb01G022450.1* and *Poalb09G003560.1*) were also found, which were commonly used as marker genes of cell division. Abundant genes involved in DNA division were also found, including DNA polymerase (*Poalb10G008020.1* and *Poalb13G012160.1*), DNA replication licensing factor (*Poalb01G006160.1*), DNA helicase (*Poalb01G038990.1*) and replication proteins (*Poalb06G009500.1* and *Poalb03G010350.1*).

We analyzed transcription factors in the turquoise module in order to understand the regulatory mechanism. Total of 1943 transcription factors were predicted in the whole genome by TFDB website and turquoise module included 858 transcription factors (Appendix A). There were the factors related to the leaf cell division including GRF (*Poalb01G010210.1*, *Poalb14G001150.1* and *Poalb07G002850.1*), E2F/DP (*Poalb15G010600.1* and *Poalb16G010500.1*), MYB3R (*Poalb06G006500.1* and *Poalb18G010720.1*) and WOX (*Poalb12G003710.1*), besides the common transcription factors, including WRKY, bZIP, bHLH, NAC and MYB. Furthermore, there were also hormone-related transcription factors, including ARF in response to auxin, AP2/ERF in response to ethylene, ARR-B in response to cytokinin and GRAS in response to gibberellin. Many common transcription factors were also found in the blue module. Moreover, the genes that might inhibit the leaf cell division, such as TCP transcription factors (*Poalb01G010010.1* and *Poalb04G009630.1*), were discovered.

### 2.14. Correlation between Modules and Hormones

The turquoise module demonstrated positive correlations with various hormones, including SA, ACC, IAA, TZR, OPDA and GA4, with correlation coefficients of 0.81, 0.95, 0.97, 0.76, 0.77 and 0.92, respectively (Figure 5A). However, GA1 exhibited a negative correlation, with a coefficient of −0.89. On the other hand, the blue module showed opposite characteristics compared to the turquoise module. It displayed negative correlations with SA, ACC, IAA, TZR, OPDA and GA4, with correlation coefficients of −0.82, −0.81, −0.84, −0.91, −0.49 and −0.59, respectively. In contrast, GA1 exhibited a positive correlation with a coefficient of 0.74. Previous studies illustrated that IAA was necessary for leaf development [4]. The great distribution of IAA in PCA analysis and the finding of ARF transcription factors in the turquoise module both indicated that IAA might play an indispensable role in leaf development. We selected IAA for correlation analysis with the turquoise and blue modules, and it was evident that they exhibited strong correlations, with both *p*-value of 1 × 10^−200^ and coefficients of 0.96 and 0.86, respectively (Figure 5B,C). In the turquoise module, a great number of genes related to IAA were found including *YUCCA* (*Poalb02G018850.1*), *PIN* (*Poalb01G038050.1* and *Poalb01G038250.1*), *AUX1* (*Poalb04G014300.1* and *Poalb09G011590.1*) and *AUX/IAA* (*Poalb03G004540.1* and *Poalb01G017180.1*), whose functions were auxin production, auxin efflux, auxin influx and auxin signaling repressor, respectively. Furthermore, auxin responsive protein *SAUR* (*Poalb04G013800.1*, *Poalb06G009730.1* and *Poalb01G038390.1*) were found. Moreover, *ARGOS* (*Poalb07G002120.1*) was also detected, which could response to auxin and increase the cell number [27]. *Poalb10G008680.1*, *Poalb10G007720.1* and *Poalb08G009890.1* were identified as TMK kinases which could be activated by auxin and phosphorylate the H^+^-ATPases [28]. In the blue module, GH3 (*Poalb02G010580.1* and *Poalb14G007690.1*) might catalyze IAA degradation [29]. AUX/IAA would be degraded by TIR protein (*Poalb01G028310.1*) [30]. As the auxin receptor for TMK-based cell-surface signaling, ABP (*Poalb06G022290.1*) mediated the global phosphorylation and auxin canalization [31]. Therefore, the close relationships between two modules and auxin revealed that the young leaves were sensitive to auxin.

### 2.15. Cyclins and CDKs in the Turquoise Module

In the turquoise module, *cyclins* and *CDKs*, which had significant expression changes, were selected and *cis*-acting elements were analyzed (Figure 6). Ten *cyclins* and three *CDK* members contained auxin response elements. In addition, GA-related elements, SA-related elements and division-related elements had also been shown. The genes containing G-box (CACGTG) element might be regulated by bZIP and bHLH transcription factors, while the genes containing W-box (TTGACC) element might be controlled by WRKY transcription factors [32,33,34]. In *P. alba*, the expression level of the *CDKA* including few *cis*-acting elements did not change significantly.

### 2.16. Interaction between Co-Expressed Cyclins and CDKs by Y2H

Previous studies found that CDKA could interact with CYCD to promote the G1-to-S phase, while CDKB-CYCB complex could activate the G2-to-M phase [35]. On the premise of co-expression and the auxin response *cis*-acting element (Figure 6), the PoalbCDKA;1-PoalbCYCD1;4, PoalbCDKA;1-PoalbCYCD3;3, PoalbCDKA;1-PoalbCYCD3;5, PoalbCDKB2;2-PoalbCYCB1;1 and PoalbCDKB2;2-PoalbCYCB2;2 were selected to identify their interactions by Y2H (Figure 7). As a HIS3 inhibitor, 3-AT was added to the medium. During the 6 days observation, PoalbCDKA;1-PoalbCYCD1;4, PoalbCDKA;1-PoalbCYCD3;3 and PoalbCDKA;1-PoalbCYCD3;5 could grow on the DDO, QDO, QDO + 10 mM 3-AT and QDO + 30 mM 3-AT medium, while PoalbCDKB2;2-PoalbCYCB1;1 and PoalbCDKB2;2-PoalbCYCB2;2 failed to develop. The growth condition on the QDO + 3-AT medium revealed that the interaction strength of the PoalbCDKA;1 with PoalbCYCD had the following relationships: PoalbCYCD1;4 > PoalbCYCD3;5 > PoalbCYCD3;3.

### 2.17. The Verification of Transcriptomic Effectiveness by qRT-PCR

To certificate the availability of transcriptomes, the expression patterns of seven genes in Y2H experiment were picked up to be determined by qRT-PCR (Appendix A). A phenomenon that each gene showed the highest expression level in the L1 sample appeared in both the results of qRT-PCR and transcriptomes. The expression of *PoalbCDKA;1* displayed the parallel level between the qRT-PCR and RNA-seq data with slightly downward trend. Although insignificant differences were detected, declining trends were noted in the expressions of *PoalbCDKB2;2*, *PoalbCYCB1;1* and *PoalbCYCB2;2*. Three *PoalbCYCDs* revealed the lowest expression in L5, and the same situation in the transcriptome presented. These genes showed the parallel expression profiles with the transcriptomes, which verified the effectiveness of the transcriptomes.

## 3. Discussion

In the CDK gene family identification, due to the difficulty of distinguishing the CDK and CKL sequences, 18 CDK members in *P. trichocarpa*, 14 CDK members and 15 CKL members in *A. thaliana* provided important references [15,36]. CYCD should be paid attention. This group contained numerous members and previous studies revealed that functional redundancy was likely to exist [35]. In the conserved motif analysis, the motif 9 was only in the D-type cyclins, which suggested that they might serve the special functions. Furthermore, 17 duplicated gene pairs in CYCD were found and each of the five CYCD3 was collinear with the other four genes, which indicated that this group might play an important role during the regulation of cell division. In the promoter analysis, many *cis*-acting elements related to hormones, plant growth, abiotic and biotic stress were discovered on the promoters of *cyclins* and *CDKs*, which manifested that they could be regulated by various plant hormones, involved in plant development and responded to the abiotic and biotic stress. In the interaction prediction, PoalbCDKA;1 was considered to play a critical role because of the central site of the network. Previous studies showed that CDKC-CYCT complex could phosphorylate RBR protein and the RNA polymerase II [37,38]. Moreover, a small system of PoalbCDKC and PoalbCYCT might possess a similar function.

In the first stage of the experiment, the rising expressions of genes related to cell wall softening, histone acetylation and division indicated that the cell wall might be loosened, the transcriptional activity might be improved and the overall activity of cell division might increase, respectively [39,40,41,42]. The upregulation of *GH3* can regulate auxin homeostasis by catalyzing the combination of IAA and amino acids, which leads to auxin degradation [29]. The *ABP* and *TMK* were both downregulated. This evidence suggested that the sensitivity of leaf cells to auxin might decrease and the auxin signaling pathway might be weakened. The increased expressions of genes related to DNA replication indicated that the cells might be in the S phase. After passing the checkpoint of the G1-to-S phase, CYCD would be degraded [35]. Moreover, because of the downregulation of *PoalbCYCD1;1*, it was speculated that the original leaf cells might be mostly in G1 phase. The expressions of some genes, including *GRF*, *ARGOS*, DNA helicase and S-phase related F-box proteins, were all enhanced. Consequently, we speculated that leaf cells might become vibrant after auxin stimulation and cross the checkpoint of the G1-to-S phase.

In the second stage of the experiment, the data of leaf area, hormones and transcriptomes were used for PCA and WGCNA analyses. Since the turquoise module was considered to be positively correlated with young leaves, it was speculated that young leaves were highly vigorous when analyzing the GO and KEGG enrichment. The DNA replication and cell division occurred commonly, and proteins were synthesized and modified in large quantities with active metabolism. Moreover, the blue module was negatively correlated with young leaves, which meant that the young leaves might be hard to resist external stress, and the ubiquitination might be inhibited in order to maintain the vibrant life activities.

The activation of *KNOLLE* transcription was positively regulated by MYB3R transcription factors in *A. thaliana* [43,44]. The *PCNA* was always regauged as the S-phase-specific proliferating cell nuclear antigen [45]. A finding that *KNOLLE* and *MYB3R* were both in the turquoise module evidenced that the young leaves were in the energetic state of cell division. Due to the discovery of a great number of hormone-related transcription factors in the turquoise module, the young leaves might be regulated by various hormones. The function of TCP transcription factor was repressing the activity of marginal meristem and accelerating the switch from cell proliferation to cell differentiation [46]. Its appearance in the blue module further verified that the cells in mature leaves might be in the expansion stage.

Many studies have reported that enhancing auxin signaling pathway can activate cell division, but the molecular mechanism remains to be investigated [4]. Our study found that the turquoise module was positively correlated with both leaf development and IAA, and contained numerous genes related to cell division and the auxin signaling pathway. Consequently, we speculated that auxin influenced *cyclins* and *CDKs* through its pathway and ultimately led to cell division (Figure 8). YUCCA, PIN and AUX were responsible for the auxin production, efflux and influx, respectively. Inside the cell, after being captured by the TIR1, auxin enhanced the combination between TIR1 and AUX/IAA [47]. Then, AUX/IAA would be degraded and ARF transcription factor would be released. Two *cis*-acting elements including AUXREs (TGTCTC or TGTCGG) and TGA-element (AACGAC) were both recognized by ARF [48,49,50]. AuxRR-core (GGTCCAT) could also be a response to auxin based on the prediction of PlantCARE website. ARF and *cis*-acting elements might be the critical keys to the connection of auxin signaling pathway and cell division. The *cyclins* and *CDKs* with these elements might be affected. During the G1-to-S phase transition, D-type cyclins form complexes with CDKA, which are then activated by CDK-activating kinase (CAK). Subsequently, retinoblastoma-related protein (RBR) is phosphorylated by these complexes and E2F/DP transcription factors are released, which can motivate the expression of S-phase-related genes [35]. During the G2-to-M phase transition, formation and phosphorylation of CYC–CDK complexes both occurred again. MYB3R transcription factors are activated by these compounds and promote the expression of M-phase-related genes [43]. The Y2H experiment revealed that PoalbCDKA;1 could interact with three members of PoalbCYCD, but PoalbCDKB2;2 did not show interaction relationships with PoalbCYCB1;1 and PoalbCYCB2;2. Consequently, based on the co-expression analysis and the result of Y2H, PoalbCYCD1;4, PoalbCYCD3;3 and PoalbCYCD3;5 might interact with PoalbCDKA;1, which could be the critical factors to activate the G1-to-S phase transition.

After being activated by auxin, ABP-TMK component phosphorylated the H^+^-ATPase, which caused hydrogen ion to flow out of the cell [31]. Finally, cell wall acidification and cell elongation occurred successively [28]. Although ABP belonged to the blue module, the expression pattern was decreasing from the first leaf to the fifth leaf. Its eigengene connectivity was −0.95 in the blue module but 0.90 in the turquoise module (Appendix A). Moreover, TIR shared the similar situation. Consequently, these two genes were positively related to the leaf development.

In the turquoise module, a total of 78 *bHLH* genes were identified. Among them, two specific *bHLH* genes in *A. thaliana*, potentially regulated by *ARF5*, have been observed [51]. They can form a heterodimer to activate transcription of *LONELY GUY 4* (*LOG4*), which can promote the periclinal division by stimulating the CK synthesis [52,53,54]. Two *bHLH* genes (*Poalb02G011550.1* and *Poalb15G000390.1*) with extremely high eigengene connectivity both reaching to 0.996, might be affected by auxin and play key roles in leaf development. Moreover, it was worth noting that the ARF, E2F/DP, GRF and MYB3R transcription factors have been associated with auxin and division. These factors might influence each other and affect the *cyclins*, *CDKs*, *KNOLLE*, *PCNA*, *ARGOS* and the genes related to DNA polymerase, replication and helicase through the *cis*-acting elements.

Leaf originates from primordium which occurs in the area of high auxin concentration in the shoot apex meristem [4]. The cells in young leaves are all in the stage of division. As a leaf grows up, the “arrest front” appears on the tip and moves to the bottom gradually [7,8]. In the WGCNA analysis, the genes expression in the turquoise module declined gradually with the development of leaves. There were many marker genes related to the proliferation, including *KNOLLE*, *PCNA*, *DNA polymerase*, *GRF*, *WOX*, *cyclins* and *CDKs*. The growth rate of the leaf area gradually slowed down from young leaves to mature leaves. Therefore, we thought that the proportion of leaf cells in the state of division was gradually reduced, and the “arrest front” might move from the tip to the bottom of the leaf with the gradual maturation. According to the pattern of leaf development, we conjectured that on the 6-month-old poplar that is about one meter high, most cells in the first leaf were in a vigorous stage of cell division when the area of the apical first leaf reached about 800 mm^2^ mid-August (Figure 9). When the area of the second leaf reached about 1900 mm^2^, cell division might begin to be inhibited. Moreover, the area of the third leaf reached about 3000 mm^2^, the inhibition of leaf cell division would be further aggravated. When the leaf area of the fourth leaf reached 3700 mm^2^, the leaf cell division basically stopped. When the leaf area of the fifth leaf was about 4300 mm^2^, the leaf cells might be completely in a state of differentiation and expansion.

Various hormones may affect the leaf development. Gibberellin (GA) inhibited the expression of *KIP-RELATED PROTEIN 2* (*KRP2*) and *SIAMESE* (*SIM*), which resulted in the leaf cells proliferation and leaf growth [55]. When brassinosteroid (BR) signaling pathway was suppressed, a smaller leaf and a lower expression of *CYCB* both appeared [56]. ERF4 involving in the ethylene signaling pathway directedly inhibited the expression of *CYCA2* to promote the endoreduplication [57]. Cytokinin (CK) can affect the meristem of leaf margin and leaf type [58]. During leaf cell expansion, cytokinin were responsible for cell wall elongation, turgor pressure increases and endoreduplication [59]. In our research, GA4 showed the same trend with IAA, but GA1 performed the rising trend. SA contributed to the immunity against biotrophic pathogens, while JA increased the resistance against necrotrophic pathogens [60]. The decreasing trend of SA was found and JA changed gently in our measurement. When the positive regulators of ethylene signaling pathway were mutated, larger leaves appeared [61]. However, ACC showed a tendency toward decline. There seemed to be some contradictions between the metrical results and the previous reports. We thought that these differences might be a consequence of our exogenous application of auxin. This amplification of auxin signals might interfere with other hormones, possibly as a result of intrinsic hormone self-regulation.

Leaf development involves the cell division and cell expansion. Although the turquoise module received more attention, it is worth noting that the blue module might contain numerous genes associated with leaf differentiation and warrant further investigation. An interesting phenomenon that *PoalbCYCD1;6* and *PoalbCDKG;4* appeared in the blue module with gradually increased expression remained to be investigated. TCP transcription factors (*Poalb01G010010.1* and *Poalb04G009630.1*) dominated the leaf cells differentiation. ERF transcription factors (*Poalb03G012690.1*) might mediate the cell endoreduplication. Combined with cell division and expansion by dissecting the turquoise and blue modules, leaf morphogenesis can be further analyzed. We can further analyze big data by integrating additional transcriptome data from poplar leaves and even single-cell sequencing data to construct a more comprehensive model of leaf development. This approach may provide insights into the factors contributing to the rapid growth of *P*. *alba*, as well as the molecular mechanisms of notable resistances of insect, germs, drought and cold stresses.

## 4. Materials and Methods

### 4.1. Identification of the Cyclin and CDK Gene Families in P. alba

The *P. alba* genome, *A. thaliana* genome and *P. trichocarpa* genome were obtained from the Chinese Academy of Forestry, TAIR website (https://www.arabidopsis.org/, accessed on 1 December 2021) and Phytozome website (https://phytozome-next.jgi.doe.gov/, accessed on 1 December 2021), respectively. To identify cyclin members, hidden Markov model (HMM) profiles of the cyclin_N domain (PF00134) and cyclin_C domain (PF02984) were downloaded from Pfam database (http://pfam.xfam.org/, accessed on 3 December 2021) and the file of the 49 cyclin protein sequences in *A. thaliana* after multiple sequence alignment using the MAFFT website (https://mafft.cbrc.jp/alignment/server/, accessed on 3 December 2021) was also regarded as the HMM profile, after which putative cyclin protein members in *P. alba* were initially identified by HMMER3.0 [62]. Subsequently, candidate cyclin sequences were examined by the Pfam database (http://pfam.xfam.org/, accessed on 4 December 2021) and NCBI-CDD website (https://www.ncbi.nlm.nih.gov/Structure/cdd/wrpsb.cgi, accessed on 4 December 2021). To obtain putative CDK members, a BLASTP method [63] was performed in a local protein database of *P. alba*, using known *P. trichocarpa* CDK protein sequences as queries. The Pfam database (http://pfam.xfam.org/, accessed on 4 December 2021) and NCBI-CDD website (https://www.ncbi.nlm.nih.gov/Structure/cdd/wrpsb.cgi, accessed 4 on December 2021) were used to ensure the CDK domain. The CKL sequences in *P. alba* were removed after phylogenetic analysis.

### 4.2. Phylogenetic Analysis and Classification

After multiple alignments using the MAFFT website (https://mafft.cbrc.jp/alignment/server/, accessed on 2 March 2022), 63 cyclin protein sequences in *P. alba* and 49 cyclin members in *A. thaliana* were used to calculate the parameters in phylogenetic analysis via Modelgenerator software [64]. These parameters were used in the PhyML program to build trees according to the maximum likelihood (ML) method with 1000 bootstrap replicates [65]. The final phylogenetic tree was modified by iTOL website (https://itol.embl.de/itol.cgi, accessed on 20 March 2022). The 17 CDK sequences in *P. alba*, 18 CDK sequences in *P. trichocarpa*, 14 CDK sequences and 15 CKL sequences in *A. thaliana* shared the same operations.

### 4.3. Sequences Analysis

Molecular weight and isoelectric point were calculated by ExPASY website (https://web.expasy.org/compute_pi/, accessed on 5 March 2022). Gene structure and conserved motifs were identified using GSDS website (http://gsds.gao-lab.org/index.php, accessed on 5 March 2022) and MEME website (https://meme-suite.org/meme/, accessed on 5 March 2022), respectively. When searching the conserved motifs, 10 types of motifs were predicted with a width from 6 to 50 aa. Then, the charts were modified by TBtools software v1.09876 [66].

### 4.4. Chromosomal Distribution and Duplicated Gene Pairs

Detailed information of chromosomal distribution was acquired from *P. alba* genome database. All *cyclin* and *CDK* genes were mapped onto chromosomes via TBtools [66]. After BLASTP [63], DupGen_finder was used to search for duplicated gene pairs [67]. The duplicated gene pairs of *cyclin* and *CDK* genes were exhibited by TBtools [66]. The Ka and Ks of duplicated *cyclin* and *CDK* gene pairs were calculated via KaKs_calculator 2.0 software [68].

### 4.5. Promoter Analysis

The upstream sequences (2000 bp) of *cyclin* and *CDK* genes were analyzed by PlantCARE online tool (http://bioinformatics.psb.ugent.be/webtools/plantcare/html/, accessed on 10 March 2022) to predict the *cis*-acting elements. AUXREs and TGA-element both were searched manually [48,49,50]. The *cis*-acting elements were classified into three parts involving in phytohormonal response, plant growth and development and abiotic and biotic stress.

### 4.6. Interaction Prediction of Cyclin and CDK Proteins

Cyclin and CDK protein sequences in *P. alba* were submitted to the STRING website (https://cn.string-db.org/, accessed on 25 March 2022) with *A. thaliana* as the organism species. After the blast, only experiments of active interaction sources would be selected and the minimum required interaction score showed high confidence (0.700). The disconnected nodes in the network would be removed.

### 4.7. Auxin Treatment and Leaf Samplings of Seedlings in P. alba

In the first stage of the experiment, the seedlings that were about 40 cm high were cultivated indoors at 23 °C, 50% relative humidity with 16 h of light (6000–8000 lx) and 8 h of darkness. The peat soil and perlite were mixed in equal proportions as soil for cultivation. The chassis was filled with water to keep the soil moist. The fifth leaf with an area of about 5000 mm^2^ on the plant was applied with 100 μM IBA, while the control group was treated with water. After three hours, leaves were picked up for RNA-seq.

In the second stage, the seedings that were about 1 m high were cultivated outdoors in the Beijing forestry university at 21–34 °C, 56% relative humidity with 14 h of light and 10 h of darkness. The peat soil and perlite, mixed equally, served as the cultivation soil. We watered each tree with about 1 L of water every day. In the mornings, 100 μM IBA was sprayed on the aerial parts of plants for three days in mid-August. Moreover, leaves were picked up at 72 h (the morning of the fourth day). The area of the first leaf from the apex was 800 mm^2^, and we counted down to the fifth leaf. The leaves were individually photographed on the plants and compiled for comparison. ImageJ software was used to record the area of the leaves [69]. Significant difference was calculated by the SPSS software using an ANOVA via Tukey test (*p* < 0.05). After collection, leaves were frozen and ground into powder with liquid nitrogen. The powder was divided into three parts as three biological replicates for RNA-Seq and hormone measurement.

### 4.8. Transcriptome Measurement

After extracting total RNA, the quality and quantity of RNA were detected by Nanodrop and Agilent. Qualified RNA was processed for library construction. The raw data were generated using the Illumina NovaSeq 6000 platform as PE150. After the raw reads were filtered, 5.74 GB of clean data and above the 95.24% of Q30 bases per sample were obtained in the first experiment, while 5.81 GB of clean data and above the 91.98% of Q30 bases in each sample were acquired in the second experiment of leaf development. Fastp software was used to control the quality of clean data [70]. The genomic index of *P. alba* was built and the cleaned reads were mapped to the genome via hisat2 [71]. Then, Samtools software compressed and sorted the mapped data [72]. The read counts and TPM of each gene were calculated by R script. The R packages, including DESeq2, clusterProfiler and ggplot2, were used for differential expression genes analysis and plot drawing. In the first stage experiment, the fold change > 1.5 and FDR < 0.05 were both regarded as the threshold for filtering DEGs.

### 4.9. Hormone Measurement

About 0.10 g each sample was taken into centrifuge tubes and 1 mL extraction solution was added. After extraction by methanol, supernatant was centrifuged and filtered into a brown sample bottle with a 0.22 µm micromembrane filter. The chromatographic column was T3 (2.1 × 100 mm, 1.8 µm). The flow rate of gradient elution was 0.3 mL/min with 10 µL sample size at 40 °C. The mass spectrum conditions were electron spray ionization (ESI) and positive/negative ionization mode by the Ultra-high performance liquid chromatography-triple quadrupole tandem mass spectrometer (Waters company TQ-XS, Massachusetts, USA). Multi-reaction monitoring mode (MRM) was used for scanning.

### 4.10. PCA, PCoA and WGCNA Analysis

R packages, including ade4, RColorBrewer, vegan and ggplot2, were used for the principal component analysis (PCA) of hormones and principal coordinate analysis (PCoA) of transcriptomes. WGCNA package was utilized to perform weight gene co-expression network (WGCNA) based on the TPM values and trait data. To ensure construction of the scale-free network, we calculated a soft thresholding power, which was 10. Then, a one-step method was used to construct the network and examine the modules with parameters as follows: maxBlockSize: 31,000, power: 10, minModuleSize: 30, mergeCutHeight: 0.25. The correlations between modules and samples and between modules and phenotypes were calculated and visualized with *p*-value. The eigengene connectivity was computed because the module to which a gene belongs was determined by the absolute value of it. The gene annotations of GO and KEGG were derived from the company Biomarker Technologies. Moreover, the R packages, including topGO, clusterProfiler and ggplot2, were used for further analysis. The −log10(p.adj) in the GO plot and -log10(Qvalue) in the KEGG plot were showed. The transcription factors were annotated via plant TFDB website (http://planttfdb.gao-lab.org/, accessed on 5 October 2022).

### 4.11. Interaction between Co-Expressed Cyclins and CDKs by Y2H

The coding sequences of *PoalbCDKs* and *PoalbCYCs* were cloned and ligated into pGBKT7 and pGADT7 vectors, respectively (Appendix A). The recombination vectors were co-transferred into AH109 competent cells with the yeast transformation kit (SK2400-200, Coolaber, Beijing, China). After the filtration of DDO medium and PCR analysis, the 5 µL yeasts were inoculated on DDO, QDO, QDO + 10 mM 3-AT and QDO + 30 mM 3-AT medium lasting for 6 days at 28 °C.

### 4.12. The Verification of Transcriptomic Effectiveness by qRT-PCR

After extracting the RNA from the samples via RNAprep Pure Plant Kit (Polysac-charides and Polyphenol-ics-rich, DP441, TIANGEN, Beijing, China), cDNA was synthesized using FastKing RT Kit (KR116, TIANGEN, Beijing, China). Quantitative RT-PCR was performed on the BIO RAD CFX Connect Real-Time PCR Detection System (CFX Connect, BIO-RAD, Hercules, CA, USA) with SperReal ProMix Plus (SYBR Green, FP205, TIANGEN, Beijing, China). The primers were listed in the Appendix A. The PCR program consisted of 95 °C with 15 min, followed by 40 cycles of 10 s at 95 °C, 20 s at 58 °C and 30 s at 72 °C. The ∆∆Ct method was used to calculated the relative expression level of genes with an *ACTIN* gene.

## 5. Conclusions

In general, this study systematically analyzed the *cyclins* and *CDKs* gene families and employed transcriptomes and hormone measurements to investigate the possible *cyclins* and *CDKs* involving in leaf development and the potential mechanism of cell division that might be affected by the auxin. Sixty-three cyclin members and seventeen CDK members in *P. alba* were identified. The main reason that led to the expansion of *cyclin* and *CDK* genes was WGD. PoalbCYCD1;4, PoalbCYCD3;3 and PoalbCYCD3;3 were supposed to interact with PoalbCDKA;1 to participate in leaf cell division. We proposed that this process could be regulated by the auxin signaling pathway, with the ARF transcription factor playing a key role in connecting these two processes. Additionally, the rule of leaf cell types variation was also conjectured in poplar. Our study yielded innovative insights into leaf morphogenesis and provided the big data support for the poplar leaf development patterns. The speculations we put forward can contribute to rapid and healthy poplar cultivation.

## Figures and Tables

**Figure 1 ijms-24-13445-f001:**
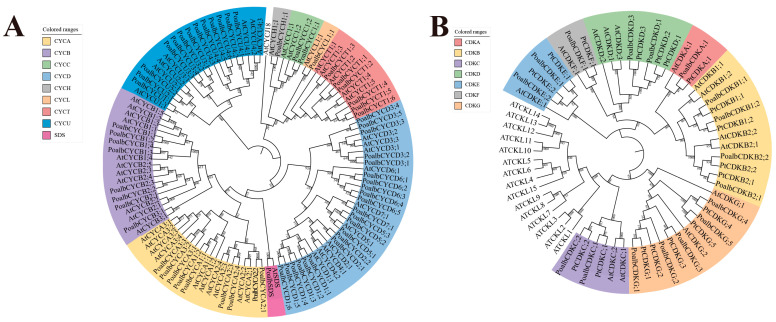
Phylogenetic trees of cyclins and CDKs. (**A**) Phylogenetic tree of cyclins in *P. alba* and *A. thaliana*. Sixty-three cyclins in *P. alba* were classified into nine types, i.e., A to D, H, L, T, U and SDS. (**B**) Phylogenetic tree of CDKs in *P. alba*, *P. trichocarpa* and *A. thaliana*. Seventeen CDKs in *P. alba* were classified into seven groups from A to G. The protein sequences were aligned by MAFFT, then the phylogenetic tree was constructed using PhyML by maximum likelihood (ML) method with 1000 bootstrap replicates.

**Figure 2 ijms-24-13445-f002:**
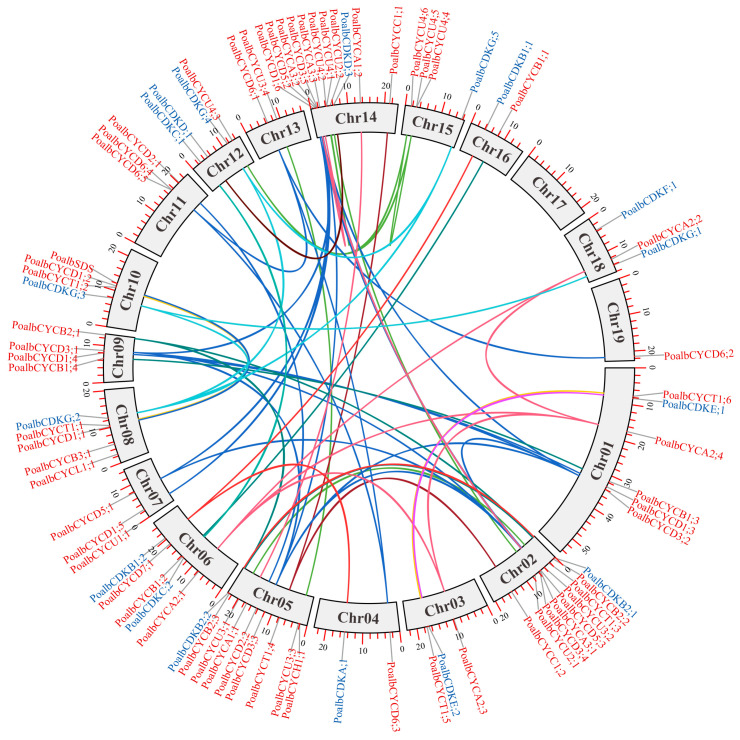
Circos figure for chromosome distribution with duplicated genes. The colorful lines represented the duplicated *cyclin* and *CDK* gene pairs. The duplicated gene pairs used the same color lines.

**Figure 3 ijms-24-13445-f003:**
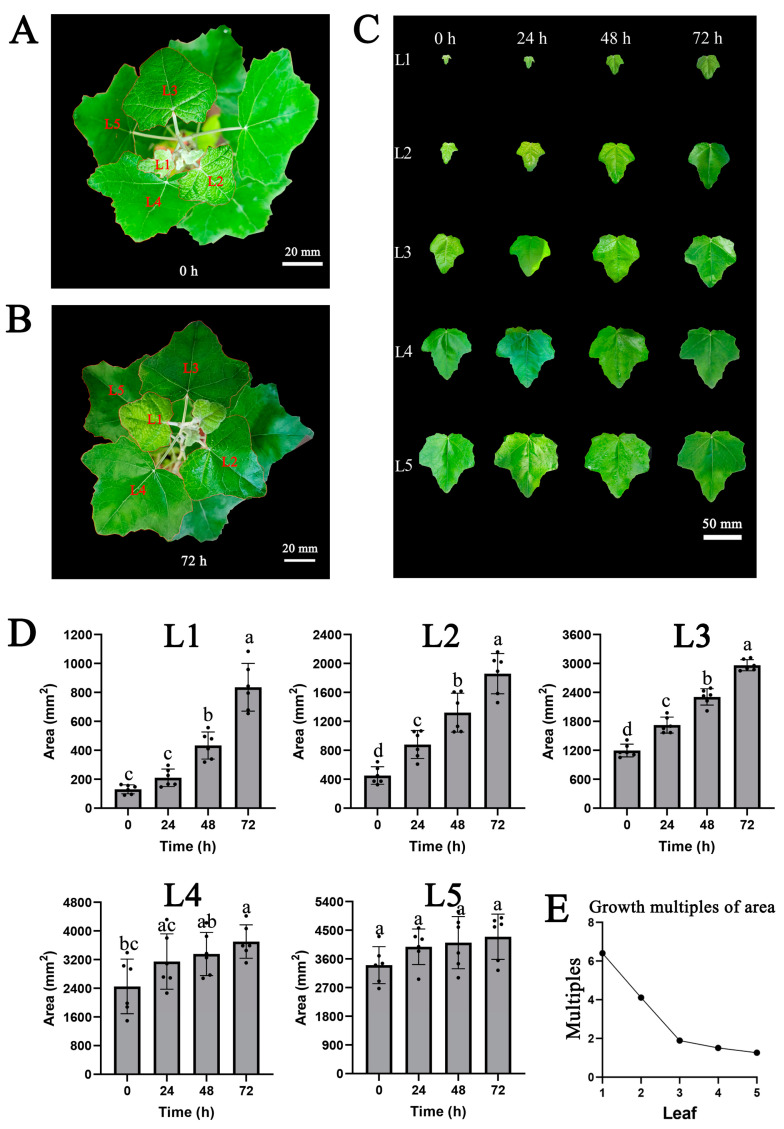
The growth curve of leaf area from young leaves to mature leaves. (**A**) The seeding at 0 h. Bar = 20 mm. (**B**) The seeding at 72 h. Bar = 20 mm. (**C**) The leaves variation under auxin stimulation lasting for three days. The leaves were individually photographed and compiled for comparison. The first to the fifth rows were the first to the fifth leaf variation, respectively. The first to the fourth columns were the photos taken at 0 h, 24 h, 48 h and 72 h, respectively. The leaves were picked at 72 h. Bar = 50 mm. (**D**) Histogram of leaves area variation. The dots represented the individual leaves area. The error bars indicated standard deviation (SD). Different lower-case letter (a–d) indicated significant difference using ANOVA by Tukey test (*p* < 0.05). The L1-L5 represented the first to the fifth leaf, respectively. (**E**) The growth multiples of area. Within 72 h, the growth multiples from the first leaf to the fifth leaf area were 6.40, 4.11, 1.89, 1.51 and 1.26, respectively.

**Figure 4 ijms-24-13445-f004:**
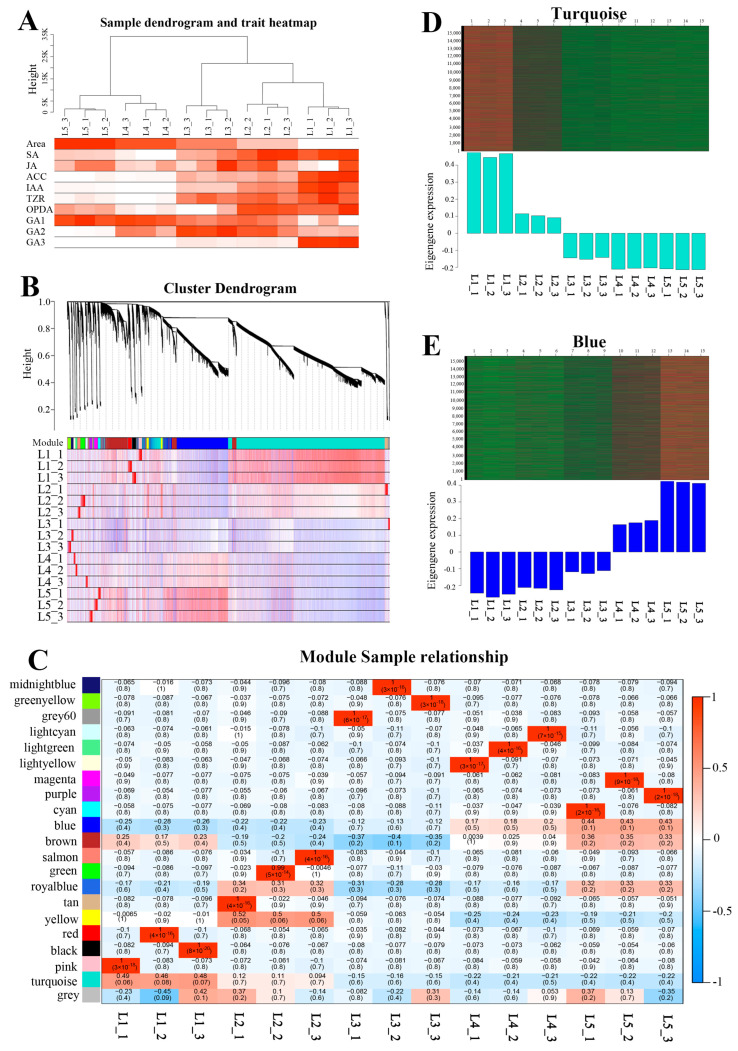
Construction of co-expression network. (**A**) Sample dendrogram and trait heatmap. The more intense the red color, the stronger the correlation between the samples and traits. (**B**) Cluster dendrogram showing modules labeled with different colors. The genes in the sample are visualized as lines, where red denotes positive correlations and blue denotes negative correlations. (**C**) Correlation diagram between modules and samples. The color key from red to blue represents values from 1 to −1. Within each box, the number above represents the correlation coefficient, while the number below represented *p*-value. (**D**) Genes expression diagram of turquoise module in the samples. The red lines represented high expression, while the green lines represented low expression. (**E**) Genes expression diagram of blue module in the samples. The meaning of lines was same as Figure 4D.

**Figure 5 ijms-24-13445-f005:**
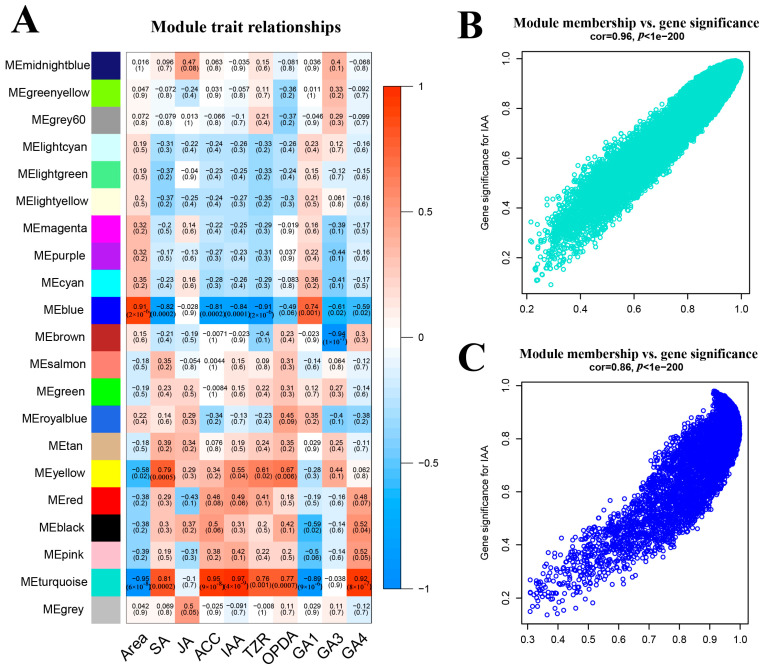
Relationships between modules and hormones. (**A**) Relationships between modules and traits. The color key from red to blue represent values from 1 to −1. Within each box, the number above represents the correlation coefficient, while the number below represents the *p*-value. (**B**) Turquoise module relationship with IAA. The coefficient of correlation was 0.96 with *p*-value of 1 × 10^−200^. (**C**) Blue module relationship with IAA. The coefficient of correlation is 0.86 with a *p*-value of 1 × 10^−200^.

**Figure 6 ijms-24-13445-f006:**
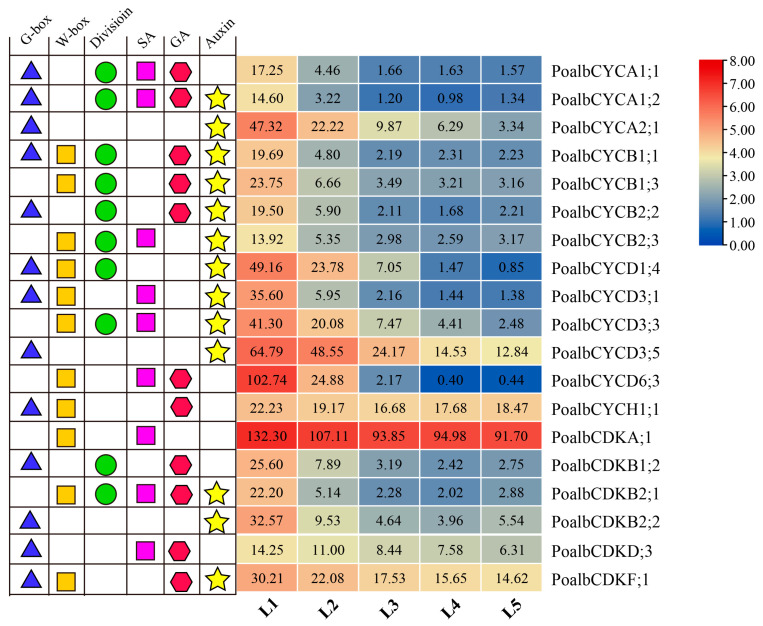
*Cyclins* and *CDKs* in the turquoise module with *cis*-acting elements. Log2 (TPM+1) was used for the heatmap. The blue triangle, orange square, green circle, purple square, red hexagon and yellow five-pointed star represented the *cis*-acting elements related to G-box, W-box, division, SA, GA and auxin, respectively.

**Figure 7 ijms-24-13445-f007:**
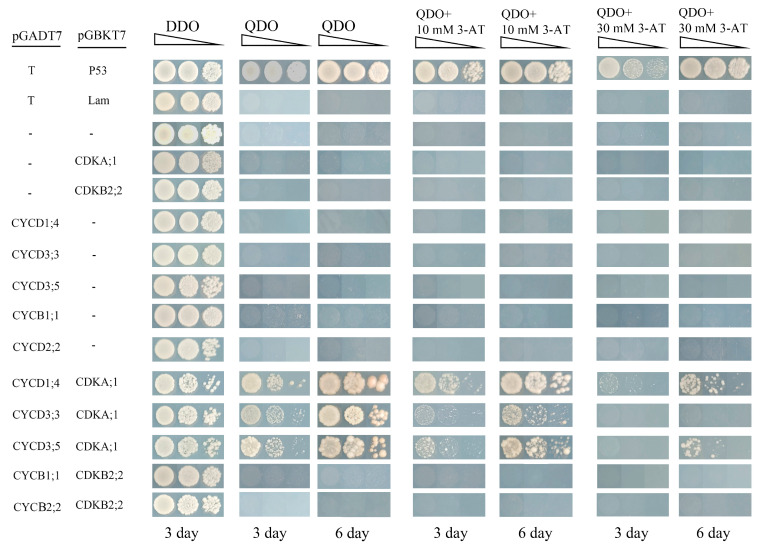
Interaction of PoalbCDKs and PoalbCYCs. The pGADT7 and pGBKT7 were constructed with the corresponding genes and co-transformed into yeast cells. The yeast cells were cultivated on the SD-Trp-Leu (DDO), SD-Trp-Leu-His-Ade (QDO), SD-Trp-Leu-His-Ade + 10 mM 3-AT (QDO + 10 mM 3-AT) and SD-Trp-Leu-His-Ade + 30 mM 3-AT (QDO + 30 mM 3-AT) lasting for six days. pGADT7-T + pGBKT7-P53 is the positive control, while pGADT7-T + pGBKT7-Lam is the negative control. Triangles represent the 0.1-fold gradient dilutions (1, 0.1 and 0.01).

**Figure 8 ijms-24-13445-f008:**
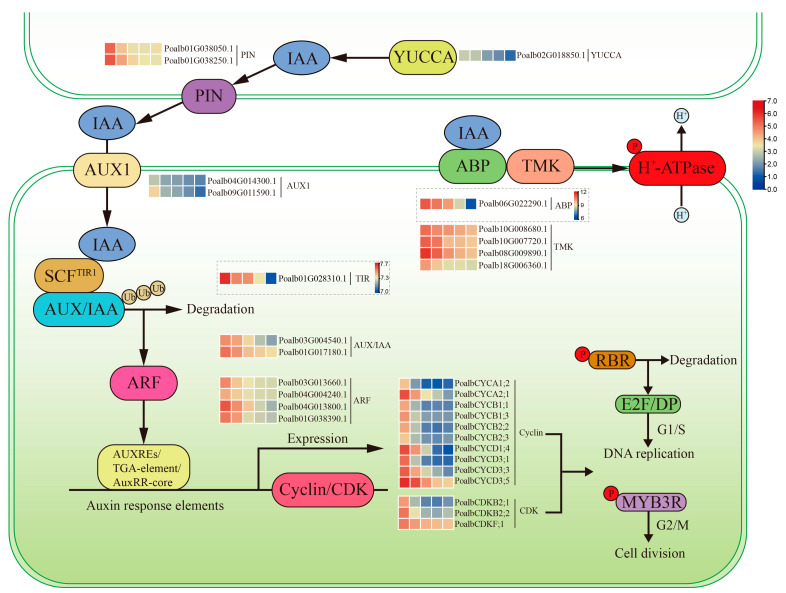
Cell division regulated by the auxin signaling pathway. Log2 (TPM+1) were used for heatmap. ABP and TIR utilized the unique heatmap scale. “Ub” was ubiquitinated protein and “P” was phosphorylation.

**Figure 9 ijms-24-13445-f009:**
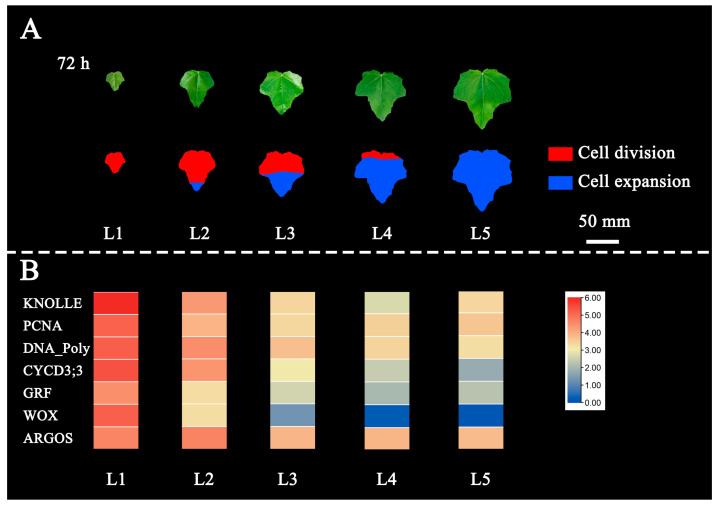
Schematic diagram of changes in the proportion of leaf cells of division and expansion. (**A**) Leaf cell distribution. The red part represents the area of cell division, while the blue part represents the area of cell expansion. (**B**) The expressions of marker genes related to cell division. Log2 (TPM+1) is used for heatmap.

## Data Availability

Raw data of transcriptomes can be obtained from NCBI website with accession number PRJNA885259 and PRJNA945583.

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
