# Peer review of "Study on the Interactions of Cyclins with CDKs Involved in Auxin Signal during Leaf Development by WGCNA in Populus alba"

_ijms, 2023, doi:10.3390/ijms241713445_

Round 1
Reviewer 1 Report
The article presents a study investigating the molecular mechanisms underlying leaf morphogenesis, with a particular focus on the role of cell division and its regulation by cyclin and cyclin-dependent kinase (CDK) complexes in response to auxin signaling in Populus alba.
The article states the objective of the study, is to investigate how cell cycle-related factors participate in the auxin signaling pathway during leaf morphogenesis. It outlines the techniques and analyses used, providing a clear overview of the study design. The study employs various techniques, such as gene family analysis, exogenous auxin stimulation, RNA-seq, and WGCNA, to comprehensively explore the molecular mechanisms underlying leaf development.
The article does not provide sufficient context or background information on the significance of studying leaf morphogenesis, cell division, cyclin, CDK, and auxin signaling in plants. This information would help readers understand the relevance and importance of the study.
Introduction should start with a brief introduction that provides background information on leaf morphogenesis, cell division, and the role of cyclin and CDK complexes in regulating the cell cycle in plants. Explain why investigating the interplay between cell cycle factors and auxin signaling is important for understanding leaf development.
The text provides only a brief overview of the findings, such as the upregulation of certain genes related to cell division after auxin treatment and the positive correlation of the turquoise module with leaf development and auxin. Further details and specific results from RNA-seq and co-expression analyses should be included to support the claims.
Also it lacks a discussion of the implications of the findings and their potential significance in the broader context of plant biology and leaf morphogenesis. Expand on the study's major findings, particularly the results of RNA-seq analysis and co-expression analyses, to provide a comprehensive understanding of the molecular mechanisms identified.
Include a discussion section where you can interpret the results in the context of previous research and propose potential mechanisms linking cell cycle regulation and auxin signaling during leaf morphogenesis.
The language can be improved. There are several syntax and grammar mistakes in the text.
Reviewer 2 Report
The manuscript entitled “Study on the Interactions of Cyclins with CDKs Involved in Exogenous Auxin Signal during Leaf Development by WGCNA” is well structured and the aim of the research work is relevant. However, there are some concerns regarding the current manuscript that authors should consider while revising their manuscript.
1. Figure 3D, increase the size and the resolution of the histogram.
2. Please explain how the expression profile by qRT-PCR analysis showed the parallel expression profiles with the transcriptomes.
3. The result section needs to be improved. The sentences are vague and not data-driven. For example; in section 2.9, the authors use terms like “downward trend”, “increasing trend” and “changed gently”. Please check throughout the manuscript.
4. Re-write the discussion more precisely and comprehensively and add the conclusion at the end of it.
5. Statistical analysis should be included in "Materials & Methods" and figures (figure 3, S7, S13).
Reviewer 3 Report
After analyzing the manuscript entitled “Study on the Interactions of Cyclins with CDKs Involved in Exogenous Auxin Signal during Leaf Development by WGCNA” by Liang et al. I make the following considerations:
The language and English of the manuscript are understandable and do not require significant grammatical correction. I don't really understand why it was necessary to investigate this in Populus alba. It would have been nice to explain it in a longer paragraph in the introduction. Also, the practical significance of the investigation. I don't see the background of the entire investigation, the underlying thought, concepts (however, there must have been some basis for such a large work). It would be worthwhile to start the introduction in a more understandable way (even the things that were missing earlier could be good for this).
The results section needs to be rewritten, currently many of its parts contain discussion-type statements and assumptions, these should be moved there. Only the pure results should be included here, without references to other studies or assumptions. I have marked some of these, but the entire text should be reviewed carefully to ensure that everything is in the correct section and any redundant information should be removed. In many cases, the software used is also indicated in the results section, in my opinion it is enough to describe them once in the materials and methods (but I leave this to the authors, it might not be so confusing).
Materials and Methods The growing conditions of P. alba, A. thaliana are not described. Greenhouse vs. open ground, watering (when, how often), planting medium, temperature, light, humidity, etc.?
L79-80, L92-93, L139-140, L165-166, L194-199, L204-207, L230-234, L244-246, L254-255, L256-257 ect.: assumptions, or simply do not belong in this section. These should rather be included in the discussion, and they should be justified with references (some of them are found, others are missing, but overall it would not hurt to give a little more detail about the underlying idea here).
Fig 1: the inscriptions and names of the phylogenetic trees are difficult to read. A larger figure can be used for better visibility.
L294: the term 'in vivo' should be written in italics
Fig 3D, E: the diagrams are too small, the captions are barely legible.
Fig 4: Are you sure you need all the figures? Also, they are very small.
ImageJ software reference is missing (L595). Schindelin, J. et al. Fiji: an open-source platform for biological-image analysis. Nat. Methods 9, 676; https://doi.org/10.1038/nmeth.2019 (2012).
For the reasons mentioned above, the manuscript needs a strong major revision.
The language and English of the manuscript are understandable and do not require significant grammatical correction.
Reviewer 4 Report
A very large and important work has been done to study the role of cell division in leaf morphology and the mechanisms of its regulation. Cyclin (CYC) and cyclin-dependent kinase (CDK) complexes play a major role in this process. Sixty-three members of cyclins and 17 members of CDKs in Populus alba were identified and systematically analyzed. A complex work has been carried out. Leaf plate growth from leaf 1 to leaf 5, the effect of exogenous auxin, the content of most known phytohormones, and transcriptomes were analyzed simultaneously. This approach required sophisticated bioinformatics work. Using the co-expression and Y2H assays, PoalbCYCD1;4, PoalbCYCD3;3 and PoalbCYCD3;5 were supposed to interact with PoalbCDKA;1, which could be the trigger to promote the G1-to-S phase transition. The authors suggest that auxin signaling genes may influence the cell cycle by acting on CYC-CDK complexes. The manuscript is beautifully illustrated.
However, despite the high appreciation of this work, there are some comments to which I would like to receive answers from the authors.
1. There seem to be two inaccuracies in the title of the manuscript
1.1. Why is it said about exogenous auxin signaling? There can be no exogenous auxin signals. Exogenous auxin was used, but it enters plant tissues and is likely to act as an endogenous hormone, although its fate may be somewhat different.
1.2. Data obtained only on leaves of Populus alba cannot be extended to leaves of all existing plants. It would be more correct to state the species name of the plant.
2. It was previously thought that the methods of doing the work should be described in such a way that the results of the experiments can be reproduced in other laboratories. In this manuscript the methods are described very briefly and reproduction of the results is out of the question. I cannot disagree that a detailed description of transcriptome acquisition or Y2H analysis would take up too much space, but some details of the experiments are absolutely necessary.
2.1. The conditions of plant cultivation are not described at all. This is especially strange because the authors create and analyze transcriptomes and determine the level of phytohormones, and these parameters are very strongly influenced by external factors. It is possible to grow plants in such conditions that almost all experimental results given in the manuscript will be quite different. It is necessary to specify in what soil poplar was grown, what are the lighting conditions, duration of the light period, light quality, light intensity, temperature.
2.2. From the description it is impossible to understand how the leaves were treated with auxin solution. Moreover, it does not say whether the leaves were treated on the plant or after separation from the plant. Which leaf did the authors consider to be the first?
3. In a number of cases, the authors do not specify the manufacturers of reagents and instruments. As far as I know, MDPI journals require such information.
4. Given the artifacts that often occur when using Y2H assays to study protein interactions, do the authors think that it would be useful to use other approaches to confirm the interaction of PoalbCYCD1;4, PoalbCYCD3;3 and PoalbCYCD3;5 with PoalbCDKA;1,
Round 2
Reviewer 1 Report
The article provides a glimpse into a comprehensive study on the molecular mechanisms underlying leaf morphogenesis, integrating cellular processes such as cell division, cyclin-CDK complexes, and auxin signaling pathways. Further elaboration on experimental procedures, results, and implications would help provide a more detailed understanding of the study's significance and potential applications.
Could you provide more details about the specific methods used in the gene family analysis and RNA-seq?
What was the rationale for choosing Populus alba as the study organism? Were there any specific characteristics that made it suitable for this research?
What are the potential implications of the study's findings for understanding and potentially manipulating leaf morphogenesis for agricultural or ecological purposes?
Can you elaborate on the co-expression analysis and Y2H experiment that led to the identification of PoalbCYCD1;4, PoalbCYCD3;3, PoalbCYCD3;5, and PoalbCDKA;1 interactions?
English is OK.
Author Response
Please see the attachment.
Response to Reviewer 1 Comments
Point 1: The article does not provide sufficient context or background information on the significance of studying leaf morphogenesis, cell division, cyclin, CDK, and auxin signaling in plants. This information would help readers understand the relevance and importance of the study. Introduction should start with a brief introduction that provides background information on leaf morphogenesis, cell division, and the role of cyclin and CDK complexes in regulating the cell cycle in plants. Explain why investigating the interplay between cell cycle factors and auxin signaling is important for understanding leaf development.
Response 1: In the introduction, we added the description of leaf morphogenesis and the importance of research relationship among cell cycle factors, auxin and leaf development. For example: the first and fifth paragraphs in the introduction. Line 30-36 and Line 68-73.
Point 2: The text provides only a brief overview of the findings, such as the upregulation of certain genes related to cell division after auxin treatment and the positive correlation of the turquoise module with leaf development and auxin. Further details and specific results from RNA-seq and co-expression analyses should be included to support the claims.
Response 2: In 2.7 section (Line 215-223, Line 226-228 and Line 236-244), we made modifications to the descriptions of GO and KEGG analysis. Additionally, we included the fold changes in RNA-seq for some genes in the descriptions to support our findings. In 2.11 and 2.14 section (Line 308-314, Line 319-321 and Line 365-372), we added the data of correlation coefficient and p-value to support the analysis.
Point 3: Also it lacks a discussion of the implications of the findings and their potential significance in the broader context of plant biology and leaf morphogenesis. Expand on the study's major findings, particularly the results of RNA-seq analysis and co-expression analyses, to provide a comprehensive understanding of the molecular mechanisms identified.
Response 3: The assumptions in the result section were removed to the discussion part. The four paragraphs were added to the discussion part, which described the gene family, the genes in the first stage experiment, the GO and KEGG analyses in the turquoise and blue modules, the genes in the turquoise and blue modules (Line 443-492). The description of relationship among turquoise module and leaf development and auxin was increased in discussion. Additionally, we added the description of “arrest front” of leaf development to support the conjecture of figure 9 (Line 537-540 and Line 544-546).
Point 4: Include a discussion section where you can interpret the results in the context of previous research and propose potential mechanisms linking cell cycle regulation and auxin signaling during leaf morphogenesis.
Response 4: We emphasized the role of ARF transcription factors and cis-acting elements in connecting the signaling of auxin and the cell cycle in Line 504-506. We also rewrote the description of bHLH, ARF and other transcription factors to highlight the importance of them in connecting the cell division and auxin signaling pathway in Line 523-532.
Point 5: The language can be improved. There are several syntax and grammar mistakes in the text.
Response 5: We checked the whole article and corrected the sentences with improper grammar.

Reviewer 2 Report
I am satisfied with the author's responses, but unfortunately, those changes are not reflected in the updated manuscript version. Please take a look at the file I've attached.

Reviewer 3 Report
Dear Authors,
In terms of certain parts of the Manuscript, it has improved a lot compared to the first version, but there are still some objectionable parts. These should be further improved and corrected by the Authors.
The manuscript often consists of complex sentences, which can make it difficult to read and understand. The sentences could be written more clearly and briefly. Some sentences may have phrasing and grammatical errors that need to be corrected.
I am still missing from the introduction the more general context/background that would show why it is important to study the role of cyclins and CDKs in plant development. For example, general plant development processes or the role of auxin in growth regulation could be mentioned.
There is much technical detail in the Introduction, for example detailing the mechanisms of action of the CDK and CYCD complexes. These details could be detailed later in the Methods or Results section. Also, at the end of the Introduction, the conclusions appear somewhat, which are usually included in the Discussion section.
It would be worthwhile to limit the Introduction to the research questions and hypotheses, and then summarize the expected results in the end of the Introduction. The Introduction does not clearly define the objectives of the research. What is the specific question or problem that the research seeks to answer? This should be explained in the Introduction.
The Discussion does not mention the possibilities of the research results in future research or further development of the field. Possible further research directions and the placing of the research results in a wider context would promote the value of the results. This can even be explained separately in the Conclusion section.
A kézirat gyakran összetett mondatokból áll, ami megnehezítheti az olvasást és a megértést. A mondatokat lehetne érthetÅ‘bben és rövidebben is leírni. Egyes mondatok megfogalmazási és nyelvtani hibákat tartalmazhatnak, amelyeket ki kell javítani.
The manuscript often consists of complex sentences, which can make it difficult to read and understand. The sentences could be written more clearly and briefly. Some sentences may have phrasing and grammatical errors that need to be corrected.
Round 3
Reviewer 2 Report
I am satisfied with the author's responses regarding my initial review. I recommend accepting the revised paper in its current state.
Reviewer 3 Report
The Authors made the requested changes to the manuscript, so I support its acceptance and publication.
I thank the authors for considering my suggestions.
All the best,
Reviewer3.
A SzerzÅ‘k a kért változtatásokat elvégezték a kéziraton, így annak elfogadását és közzétételét támogatom.